# Surface Heparinization of a Magnesium-Based Alloy: A Comparison Study of Aminopropyltriethoxysilane (APTES) and Polyamidoamine (PAMAM) Dendrimers

**DOI:** 10.3390/jfb13040296

**Published:** 2022-12-13

**Authors:** Masoumeh Ebrahimi, Atefeh Solouk, Ali Davoodi, Somaye Akbari, Masoumeh Haghbin Nazarpak, Alireza Nouri

**Affiliations:** 1Biomedical Engineering Department, Amirkabir University of Technology (Polytechnic Tehran), Tehran 158754413, Iran; 2Materials and Metallurgical Engineering Department, Faculty of Engineering, Ferdowsi University of Mashhad, Mashhad 9177948974, Iran; 3Sichuan University-Pittsburgh Institute (SCUPI), Chengdu 610207, China; 4Textile Engineering Department, Amirkabir University of Technology (Polytechnic Tehran), Tehran 158754413, Iran; 5New Technologies Research Center (NTRC), Amirkabir University of Technology (Polytechnic Tehran), Tehran 158754413, Iran

**Keywords:** vascular stent, AZ31 alloy, dendrimer, heparinization, corrosion resistance, aminolyzation

## Abstract

Magnesium (Mg)-based alloys are biodegradable metallic biomaterials that show promise in minimizing the risks of permanent metallic implants. However, their clinical applications are restricted due to their rapid in vivo degradation and low surface hemocompatibilities. Surface modifications are critically important for controlling the corrosion rates of Mg-based alloys and improving their hemocompatibilities. In the present study, two heparinization methods were developed to simultaneously increase the corrosion resistance and hemocompatibility of the AZ31 Mg alloy. In the first method, the surface of the AZ31 alloy was modified by alkali–heat treatment and then aminolyzed by 3-amino propyltriethoxy silane (APTES), a self-assembly molecule, and heparin was grafted onto the aminolyzed surface. In the second method, before heparinization, polyamidoamine dendrimers (PAMAM4-4) were grafted onto the aminolyzed surface with APTES to increase the number of surface functional groups, and heparinization was subsequently performed. The presence of a peak with a wavelength of about 1560 cm^−1^ in the FTIR spectrum for the sample modified with APTES and dendrimers indicated aminolysis of the surface. The results indicated that the corrosion resistance of the Mg alloy was significantly improved as a result of the formation of a passive layer following the alkali–heat treatment. The results obtained from a potentiodynamic polarization (PDP) test showed that the corrosion current in the uncoated sample decreased from 25 µA to 3.7 µA in the alkali–heat-treated sample. The corrosion current density was reduced by 14 and 50 times in samples treated with the self-assembly molecules, APTES and dendrimers, respectively. After heparinization, the clotting time for pristine Mg was greatly improved. Clotting time increased from 480 s for the pristine Mg sample to 630 s for the APTES- and heparin-modified samples and to 715 s for the PAMAM- and heparin-modified samples. Cell culture data showed a slight improvement in the cell-supporting behavior of the modified samples.

## 1. Introduction

Magnesium (Mg) is a vital metal element in the human body that has very good biocompatibility and functional qualities and shows no signs of local or systemic toxicity [1]. Mg alloys have been widely studied as promising materials for the fabrication of biodegradable metallic implants, such as scaffolds for bone healing, fracture fixation devices, cardiovascular and nonvascular stents, etc. [2]. However, excessive corrosion of Mg alloys owing to non-uniform corrosion between the second phases and α-Mg matrices restricts their application in medical implants, particularly in physiological environments with chloride ions [2,3]. In an aqueous solution, the dissolution of magnesium proceeds by the following reaction: Mg + 2H_2_O→Mg^2+^ + 2OH^−^ + H_2_↑, but the physiological environment complicates the corrosion process [4]. The resulting corrosion leads to the over-emission of Mg, the formation of gas bubbles, and complete implant failure before recovery [5]. Thus, establishing a controllable corrosion/dissolution rate is one of the key challenges for the clinical use of Mg alloys in metallic implants [6].

Controlling Mg corrosion rates and increasing its biocompatibility can be conducted via bulk and surface modification. Surface characteristics, in particular, have a key influence on the corrosion resistances, biological compatibilities, and mechanical properties of Mg alloys [7,8]. Therefore, various surface-modification methods, such as microstructure modification, the formation of chemical conversion layers, and the construction of various coatings on substrates, have been studied to enhance the biocompatibilities and corrosion rates of Mg alloys [9]. Research aimed at finding a more efficient, simpler, and cost-effective modification method is also being undertaken [10]. One of the surface-modification strategies that improves the blood compatibilities and corrosion resistances of Mg alloys is the deposition of self-assembled molecules, which creates a single layer of molecules on the surface of an alloy [11]. Silanes, phosphoric acid, carboxylic acid, and dopamine are biological molecules commonly used for cardiovascular applications [12]. Amongst them, silanes have received increased attention in recent years. Silanes are silicon-based minerals with the formula R′(CH_2_) nSi(OR)_3_, where R′ represents an organic group and R represents a hydrolyzable alkoxy group. When silane reacts with water, silanol groups -Si(OH)_3_ are formed, which connect with hydrated metal surfaces (metal-OH) through Si-O-metal chemical bonding. Meanwhile, silanol groups generate siloxane (Si-O-Si) linkages, which act as a corrosion barrier [2]. The effects of the presence of several silane groups on the corrosion characteristics of the AZ31 alloy were investigated by Brusciotti et al. [13]. The coated samples were subjected to corrosion tests by electrochemical impedance spectroscopy while immersed in 0.05 M NaCl for 1 month. The results showed that the four proposed hybrid coatings had higher corrosion resistances than those reported in the literature, with a considerable improvement shown by the APTMS- and APTES-based ones, as no relevant degradation was observed after one month of immersion. The APTES coating also showed excellent adhesion to the substrate [13]. Other studies were conducted to immobilize polyethylene [14], polymethylmethacrylate [15], and poly(lactic-co-glycolic acid) [2] on the surface of an Mg alloy using an APTMS intermediate, and the effects of different layers on corrosion resistance and blood compatibility were investigated.

Dendrimers are highly branched, star-shaped macromolecules of nanometer size [16] and are mostly studied as frameworks and carrier systems [17]. Among existing dendrimers, polyamidoamine (PAMAM) is the most widely studied dendrimer [18,19] and the most attractive carrier system for this study due to its NH_2_ functional group, its ease of preparation and functionalization, and its suitability to be used as a corrosion inhibitor [20].

Heparin is a polysaccharide substance that is highly compatible with blood and is widely used to alter the surfaces of biomaterials in contact with blood. Heparin can not only enhance the blood compatibility of materials, but also promote the growth of endothelial cells to some extent, even selectively promote the growth of endothelial cells [21]. Therefore, we further immobilized heparin on the alloy surfaces via covalent bonding between the amines of the dendrimers or APTES and the carboxyls of heparin [22].

The aim of this study was to investigate the effect of PAMAM dendrimers on the surface aminolyzation of the AZ31 alloy. The resulting heparinized surfaces were compared with the APTES-aminolyzed surfaces with respect to both corrosion resistance and blood compatibility [10,23]. Since PAMAM dendrimers have more NH_2_ functional groups on their molecular surfaces and higher drug-loading capacities than silanes (present in the APTES structure), they were effectively employed for the surface aminolyzation. The PAMAM structure leads to the entrapment of drugs between its branches. These two characteristics improve and increase heparinization. Additionally, the dendrimers might offer greater corrosion resistance than silanes due to the fact that their geometric and chemical structure serves as a corrosion inhibitor. To the best of our knowledge, this is the first work to compare the corrosion resistance and blood compatibility of a heparinized layer using both PAMAM and APTES.

## 2. Materials and Methods

### 2.1. Materials

All reagents were used as received. Sodium hydroxide (NaOH), (3-Aminopropyl) triethoxysilane (APTES), succinic anhydride, poly(amidoamin) (PAMAM4-4), heparin, ethanol, and Dimethylformamide (DMF) were purchased from Merck, Germany. The substrate was an AZ31 Mg alloy sheet with a nominal bulk content of 3% Al, 0.9% Zn, 0.334% Mn, 0.02% Si, 0.005% Fe, 0.05% Cu, 0.04% Ca, and 0.1% Be, Mg balanced. Hank’s solution containing 8 g NaCl, 0.35 g NaHCO_3_, 0.426 g Na_2_HPO_4_, 0.4 g KCl, 0.06 g K_2_HPO_4_.3H_2_O, 0.1 g MgCl_2_-6H_2_O, 0.14 g CaCl_2_, 1 g glucose, and 0.072 g Mg_2_SO_4_-7H_2_O in 1000 mL Milli-Q water was purchased from Merck, Germany.

### 2.2. Self-Assembly and Immobilization of Biomolecules 

AZ31 sheets with a thickness of 2 mm were cut into 25 × 25 mm pieces. Silicon carbide papers of 400 to 2000 grit were used to polish the pieces. Subsequently, the pieces were ultrasonically cleaned for 10 min with ethanol. The samples were air-dried before being kept in anhydrous ethanol for future use. Before silane adhesion, the ultrasonicated specimens were submerged in a 3 M NaOH solution with a pH of 12 for 24 h at 75 °C in a water bath to generate a homogeneous hydroxide coating on the specimen surfaces (M-H sample). The purpose of this test was to enhance the adhesion of the silane substrate to the AZ31 alloy and to improve its corrosion resistance. In order to silanize the substrate, the hydroxidized-surface sample was dipped into the solution of APTES (2.5%V, ethanol). Subsequently, the PAMAM was grafted onto the surface of the silanized sample. First, the samples were immersed in succinic anhydride solution (12 gr/L, DMF) for 48 h at 90 °C in an oil bath and then immersed in PAMAM solution (25 gr/L, Milli-Q water) for 12 h. After each immersion step, the samples were washed three times using the same step solvent.

Two types of samples, including M-H-A and M-H-A-De, were selected for grafting heparin onto their surfaces. Therefore, the mentioned samples were placed in a heparin sodium solution at room temperature for 5 h. The schematic structures of the self-assembled molecules on the sample surfaces are shown in Figure 1. Table 1 also briefly shows the name of the samples, surface modification, and functional group of the surface.

### 2.3. Surface Characterizations

The changes in the surface chemical structure of the AZ31 alloys following surface modification were initially studied using Fourier transform infrared spectroscopy–attenuated total reflectance (Shimadzu IRPrestige-21, ATR-FTIR spectrophotometer). At room temperature and under ambient conditions, infrared adsorption between 4000 and 650 cm^−1^ was measured. The surface morphologies of the samples following surface modification and polarization corrosion testing were observed using scanning electron microscopy (SEM) with a Vega-II XMU (Tescan Inc., Cranberry Township, PA, USA). Water contact angles were used to determine surface hydrophobicities/hydrophilicities at ambient temperature and at standard atmospheric pressure. A liquid drop photograph was obtained soon after 10 µL of distilled water was dropped on the substrate. The water contact angles were obtained using ImageJ^®^ software. 

The amine concentrations on the surfaces were confirmed using the acid orange (AO) test [24]. Before being washed with HCl solution (pH = 3), the samples were submerged in 500 µmol/L of AO-HCl solution (pH = 3) and shaken for 5 h at 37 °C. Finally, using a microplate reader set to 485 nm, the optical density (OD) of the desorbed AO supernatant was determined. The obtained ODs were proportional to the amounts of amine on the surfaces of the samples. The amounts of heparin grafted onto the surfaces were measured using the methylene blue test. The samples were immersed in methylene blue dye solution (1 mg/mL) for 5 h, after which the ODs of the desorbed methylene blue supernatants were measured at 663 nm with a microplate reader. For each sample, the tests were performed three times.

### 2.4. Corrosion Behavior

The corrosion behavior of the samples was investigated using an electrochemical analyzer (ZIVE SP1 potentiostat). A three-electrode cell was employed, with a saturated calomel electrode (SCE) as the reference electrode, a platinum electrode, as well as the sample with a 1 cm^2^ exposed area as the working electrode. The sample was stabilized in Hank’s solution at 37 °C for 600 s. The potentiodynamic polarization (PDP) test was run at a scanning rate of 5 mV/s. Tafel extrapolation and three test replications were used to calculate the related corrosion potential (Ecorr) and current density (Icorr). The data were collected from 100 kHz to 0.1 Hz using an electrochemical impedance spectroscopy (EIS) test with a 5 mV sinusoidal perturbing signal. The impedance measurements were analyzed using EIS analyzer software.

### 2.5. Cell Culture and Morphological Study 

Human umbilical vein endothelial cells (HUVECs) were acquired from the National Cell Bank of Iran (Pasteur Institute of Iran) and cultured in DMEM/F12 with 10% FBS, 100 U/mL penicillin, and 100 µg/mL streptomycin (Sigma, St. Louis, MO, USA). After reaching confluency, the HUVEC cells were removed and cultured on the surfaces of M, M-H-A-He, M-H-A-S-De-He, and polystyrene plates as a control sample at a density of 200 × 103 cells/well on 6-well tissue culture polystyrene plates (Nunc, Roskilde, Denmark) and incubated for 24 h at 37 °C in a humidified atmosphere with 5% CO_2_ and 95% air. At this stage, the endothelial cell morphologies in contact with the controls and the modified samples were observed with an optical microscope. In order to obtain a closer view of the cell morphologies in contact with the samples, the cells were fixed with a 2.5% glutaraldehyde (GTA) solution, dehydrated with a degrading ethanol series, and coated with a gold layer by chemical vapor deposition, all of which were observed using SEM [25].

### 2.6. Cell Viability

The biocompatibilities of AZ31 and the heparin-modified sample were evaluated by an indirect cytotoxicity assay using human umbilical vein endothelial cells (HUVECs). The extracts were prepared in a humidified environment with 5% CO_2_ at 37 °C for 24 h using 8 mL Dulbecco’s Modified Eagle’s Medium (DMEM). The supernatant was removed and refrigerated at 4 °C. To facilitate adhesion, cells were cultured in 96-well culture plates at a density of 1 × 10^4^/well for 24 h before being replenished with 100 µL extract and 10 µL fetal bovine serum. After 24 h of culture, each well received 100 µL of MTT, which was cultured for 3 h, and then 100 mL of the supernatant was measured spectrophotometrically at 490 nm using a microplate reader (BioTek Elx800). The background with MTT results for the extracts was subtracted. This test was performed with three replications for each sample.

### 2.7. Clotting Time

Selected samples were warmed for 5 min in a water bath at 37 °C. A micro-syringe was used to drop fresh human whole blood (0.25 mL) from an aspirin-free adult donor (anticoagulated with trisodium citrate at a 9:1 volumetric ratio) into the middle of the samples. The formation of a solid blood clot was next verified using a needle tip after 0.02 mL of CaCl_2_ (0.2 M) was progressively added to the blood samples. Finally, the time it took for each sample to develop a fibrin clot was calculated [26]. Three replications were performed for each sample.

## 3. Results and Discussion

### 3.1. Surface Characterizations

The full FTIR spectra for the various samples after surface modifications are shown in Figure 2. No peak was detected for the uncoated Mg sample, as can be seen in Figure 2. Although Mg is oxidized rapidly when it is exposed to air or an aqueous solution, its oxidation is largely prevented in anhydride ethanol. Table 2 summarizes the peaks and bands for each sample, according to Figure 2. 

A peak at 3684 cm^−1^ was observed on the surface of the M-H sample, which can be attributed to the adsorption of the stretching vibrations of OH groups, suggesting that alkali–heat treatment can produce hydroxyl groups on the surface. The FTIR spectrum for the M-H-A sample clearly revealed peaks around 1030–1150 cm^−1^, which correspond to Si–O asymmetric stretching in –Si–O–Si–, and a peak around 1556 cm^−1^, which corresponds to protonated amino groups. The presence of the PAMAM layer is indicated by distinctive peaks, such as C=O stretching at 1613 cm^−1^ and N-H bending at 1563 cm^−1^. In the heparin-modified sample, there are peaks around 1168 cm^−1^ and 1240 cm^−1^, which correspond to the S–O and S=O bands. The peak around 2960 cm^−1^ indicates C–H bonds in the CH_3_ and CH_2_ groups and was seen in samples modified with PAMAM, APTES, and heparin [27] (Figure 2). 

In order to measure the density of the amine groups grafted onto the surface of the samples, an anionic dye, namely, acid orange 7 (AO7), was used. The reason for using the anionic dye lies in the higher absorbance of the anionic dye on the positively charged aminolyzed surface [24]. Thus, more dye absorption can indicate a greater number of positively charged amine groups. The M-H-A and M-H-A-S-De samples were selected in this test for their amine groups, and the uncoated Mg sample was selected as a control sample.

Figure 3a shows the modified samples after immersion in the dye solution. As expected, the M-H-A-S-De sample modified with PAMAM dendrimers had more color absorption than the M-H-A sample due to the higher density of amine groups in the former sample. Owing to the corrosion of the surface and the release of Mg^+^ ions, there was some adsorption on the uncoated sample. The adsorption of anionic dye is caused by the presence of a positive charge on the surface. The effects of surface corrosion on the samples can be seen in Figure 3a.

Table 3 summarizes the results for OD adsorption from the residual solution and the densities of surface amine groups. The amine group density in the sample modified with PAMAM was almost twice that of the sample modified with APTES, as was concluded from images taken of the samples (Table 3).

Methylene blue cationic dye, which interacts electrostatically with the negative sulfate groups of heparin, was also used to measure the amounts of heparin adsorbed on the surfaces. To this end, two heparinized samples were selected and tested along with two non-heparinized samples of M-H-A and M-H-A-S-De as controls.

Figure 3b shows the amounts of dye adsorption on the surfaces of the samples. The M-H-A-S-De-He sample exhibited the highest dye uptake. As mentioned earlier, the reason for this is the much higher density of terminal amine groups in the PAMAM dendrimers than in the APTES-modified sample, resulting in better heparinization. Heparin can also be trapped in voids within the PAMAM dendrimer structure, which accounts for the increased heparin uptake compared to that of the APTES-modified sample [28]. The results for OD adsorption from the residual solution and for the densities of heparin are listed in Table 3, which are fairly consistent with the results in Figure 3b.

Surface hydrophilicity has been widely used to characterize biomaterial surface wettability. In general, improved hydrophilicity may improve biomaterial biocompatibility because most human tissues contain a lot of water [14]. Figure 4 shows the static water contact angles for different samples. For the uncoated sample of the AZ31 alloy, the contact angle was about 61. A similar value has also been reported by Gao et al. [21]. As can be seen in the diagram, the static contact angle decreased from 61° to 46° for the M-H sample, which is attributed to the formation of hydrophilic hydroxide groups on the surface [29]. There are two factors that influenced the contact angle in the M-H-A sample: (i) the presence of amine groups, which leads to hydrophilicity; and (ii) the rotation of the alkane group towards the surface, which gives rise to hydrophobicity [30]. However, the resulting hydrophobicity of the sample, due to the increase in the contact angle from 46° to 68°, indicated the stronger effect of the alkane group. The contact angles for both the M-H-A-He and M-H-A-S-De-He samples were reduced from 68° to 25° and from 23° to 8°, respectively, compared to the M-H-A sample, which reductions were due to the presence of hydrophilic groups (i.e., NH_2_ and OH) in the heparin [31]. As shown in Figure 4, the presence of dendrimers in the M-H-A-S-De sample increased the number of amine functional groups and thus reduced the contact angle to 23°. 

### 3.2. In Vitro Electrochemical Tests

All electrochemical tests were performed in Hank’s solution at 37 °C. This solution contains chloride ions, which are known to be very aggressive with respect to Mg alloys, enhancing corrosion activity. Open circuit potential (OCP) curves for various samples are illustrated in Figure 5a. As can be seen, the surface potentials of the samples were generally negative with respect to the SCE reference. The OCP of the M-H sample increased compared to the uncoated sample, evolving to a more noble potential, indicating that the passive films formed on the M-H were relatively more protective than those on the uncoated AZ31 alloy [15]. The M-H-A-S-De and M-H-A samples exhibited slightly higher surface potentials compared to the other samples, which indicates the positive effect of the presence of amine functional groups on the nobility of the surfaces of these samples. After grafting heparin onto the surfaces of the samples (i.e., the heparin-modified samples) and the reaction between amino groups and heparin, fewer amine groups were in contact with the solution, which led to decreases in the samples’ surface potentials. 

For the potentiodynamic polarization (PDP) test, the samples were polarized in the range of −500 to −300 mV relative to the OCP. The PDP results are presented in Figure 5b. The values for the corrosion currents and corrosion potentials are summarized in Table 4. As reported in Table 4, in sample M the corrosion current was 25 µA, which decreased to 3.7 µA in sample M-H. This shows that the passive hydroxide layer was well-formed and had a great effect in controlling the corrosion rate. In addition, in samples modified with APTES and dendrimers, the corrosion current density decreased by 14 and 50 folds, respectively. Increased hydrophobicity due to APTES is one of the reasons for the increase in the corrosion resistance of the M-H-A sample [29]. Additionally, dendrimers, due to their geometric and chemical structure, act as corrosion inhibitors and therefore increase corrosion resistance [32].

The results of electrochemical impedance spectroscopy (EIS) testing were analyzed using EIS analyzer software and simulated up to a frequency of 0.001 Hz. Figure 6a–g show the fitted Nyquist curves obtained using the software and the equivalent circuits considered for each sample. Figure 6g shows all the simulated diagrams in one plot to compare them with each other. The numerical results obtained with the EIS analyzer software are summarized in Table 4. The Nyquist curve for sample M (uncoated Mg) is seen in Figure 6a. Two capacitive semicircles describe the Nyquist plot. The charge transfer processes are represented by the high-frequency semicircles, whereas the mass transport relaxation (Mg^+^) in the solid phase, i.e., in an aggregating layer, is represented by the medium/low-frequency semicircles [33]. 

Subsequently, in the M-H sample (Figure 6b), due to the formation of a passive hydroxide layer, the corrosion resistance of the sample surface (R1) increased significantly, which confirms the results of the polarization and SEM images. In contrast to the polarization test, which showed a reduction in corrosion currents, the corrosion resistances of both the M-H-A and M-H-A-S-De samples also decreased.

The presence of APTES and PAMAM leads to the formation of amine groups on the surfaces, which will adsorb metal ions, such as Na^+^ and Mg^2+^, after their immersion in Hank’s solution at a pH of 7.4; however, in acidic solutions these substances can be proton adsorbents [34]. The adsorption of metal ions on the surface of a sample can cause additional current and thus lead to lower corrosion resistance.

In both the M-H-A-He and M-H-A-S-De-He samples, which had heparin grafted onto their surfaces, the corrosion resistances were increased compared to the samples without heparin modification. This increase was due to the coverage of some of the surface amine groups by heparin molecules and the decrease in the densities of the amine functional groups (Table 3 and Figure 6).

SEM analysis was used to assess the surfaces of the samples following surface modification and electrochemical testing. The lines and scratches due to the polishing of the M, M-H, and M-H-A samples before the electrochemical test can be seen in Figure 7a–d. The SEM micrographs show that these lines were covered after the grafting of PAMAM and became more uniform (Figure 7e–h). It is notable that the samples did not corrode during the coating preparation process and immersion in different solutions and that the modifications occurred without damage to the surfaces. Moreover, cracks and surface damage were not seen on the samples before the electrochemical test, indicating the stability of the coating. SEM images after the corrosion tests (Figure 7e–h) qualitatively show the intensities of the surface degradation, confirming the quantitative results obtained from the PDP test. The order of the samples from the lowest to the highest rates of corrosion and surface degradation is: M-H-A-S-De < M-H-A < M-H < M. 

### 3.3. Cell Culture and Morphological Investigation 

Rapid endothelialization of intervention stents is essential for the healing of injured vascular tissues. For DES systems, rapid endothelialization is particularly important to prevent the onset of late stent thrombosis [10]. The regeneration of a layer of healthy endothelial cells on the stent surface is necessary for better hemocompatibility. Controlled corrosion rates, surface modification, and functionalization are important prerequisites for promoting endothelial cell adhesion, proliferation, and functional expression [6].

Figure 8 shows optical and SEM micrographs of the HUVECs on the unmodified AZ31 and the two heparinized samples with heparin densities of 5.22 × 10^−7^ for M-H-A-He (Figure 8b) and 6.52 × 10^−7^ for M-H-A-S-De-He (Figure 8c).

The lower cell growth and adhesion on the unmodified AZ31 alloy could be due to the release of hydrogen bubbles on its surface and also the change in pH due to the high corrosion rate, which may inhibit cell activities. In order to prepare a suitable environment for cell adhesion and proliferation, it is necessary to first control the corrosion rate [35]. Endothelial cell morphological observations showed more cell adhesion and proliferation on both of the modified samples as compared with AZ31. There was higher ECM secretion and better cell-supporting behavior for the modified samples with larger amounts of heparin as compared with the control. The positive effect of heparin on both surface hydrophilicity and cell adhesion, accordingly, is already mentioned in the literature [25]. 

### 3.4. Cell Viability

Figure 9 shows the viability of HUVECs after 24 h of contact with sample extracts. The statistical results showed that there were no significant differences between the cell viabilities of the cells that were in touch with extracts of the modified samples for 24 h and those in contact with the unmodified ones (*p*-value > 0.05). The fact that there was no significant difference between the cell viability for the modified and unmodified samples is important in that it indicates that the coating materials and methods did not add toxicity to the system. The result may be due to the short extraction time, and further studies with longer extraction times are suggested. 

### 3.5. Clotting Time Test 

In order to study the interaction of blood with the modified surfaces, the fibrin filament formation time on the surface was measured. A faster blood clot formation time shows a higher thrombogenic capacity for a material, which can cause in-stent restenosis in implanted cardiovascular stents. The results are listed in Table 5. The M-H sample exhibited the shortest clotting time. Surface modification with APTES and PAMAM also contributed to reducing the time required for clotting. It has been reported in the literature that an amine functional group with a positive charge can accelerate the time of clot formation [36], and, as expected, the heparin-coated samples had the longest clotting times, with a maximum value of 715 s shown for the M-H-A-S-De-He sample (Table 5). Heparin has an excellent anticoagulant effect, and its anticoagulant activity is mainly due to sulfate, sulfanilamide, and carboxylic acid groups. In addition, the heparin coating on the surface of the magnesium alloy endows the surface with excellent surface wettability, which can effectively prevent the absorption of fibrinogen [6].

## 4. Conclusions

In the present study, heparin was grafted onto the surface of the AZ31 alloy using APTES and PAMAM dendrimers. The aim of this study was to use PAMAM in order to increase the numbers of amine functional groups and increase the amount of heparin and investigate its effect on corrosion resistance and hemocompatibility. FTIR testing also confirmed the surface modification of the samples at each stage. In general, the modifications of the surfaces increased hydrophilicity, higher hydrophilicities being observed for the heparinized samples. In the sample modified using PAMAM dendrimers, the number of surface amine groups, and consequently the amount of heparin grafted, was higher than for the APTES sample. Another important factor in Mg-based alloys is corrosion resistance, which was significantly increased in the modified specimens compared with the uncoated alloys. The formation of a passive hydroxide layer on a sample has a great effect on reducing the corrosion current and increasing the corrosion resistance. In this study, corrosion current density decreased by 14 and 50 folds in samples modified with APTES and dendrimers, respectively. SEM images taken before and after the corrosion tests also supported the increase in corrosion resistance. Cell viability testing and cell cultures were used to analyze the hemocompatibilities of the alloy with the uncoated surface and the two heparinized samples. The heparinized dendrimers (PAMAM) grafted onto the APTES-coated hydroxidized Mg alloy (the M-H-A-S-De-He sample) had the best results in the cell culture analysis. The results obtained for clotting times to evaluate the hemocompatibilities of the stents indicated that in the heparinized samples clotting time was increased by up to 40%. Consequently, the heparinized samples using dendrimers provided promising results in terms of corrosion resistance and hemocompatibility.

## Figures and Tables

**Figure 1 jfb-13-00296-f001:**
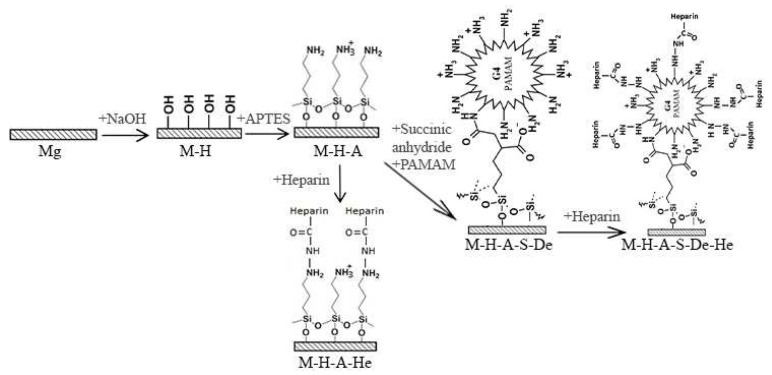
The schematic structures of the self-assembled molecules on the sample surfaces.

**Figure 2 jfb-13-00296-f002:**
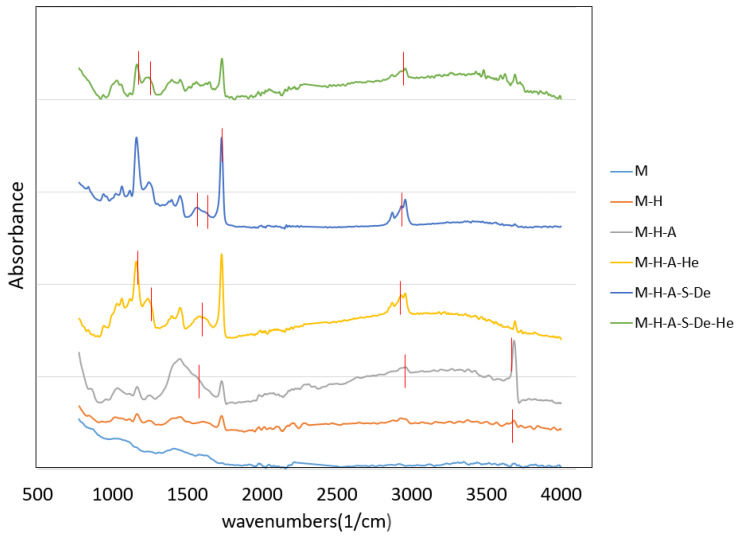
FTIR spectra for various samples.

**Figure 3 jfb-13-00296-f003:**
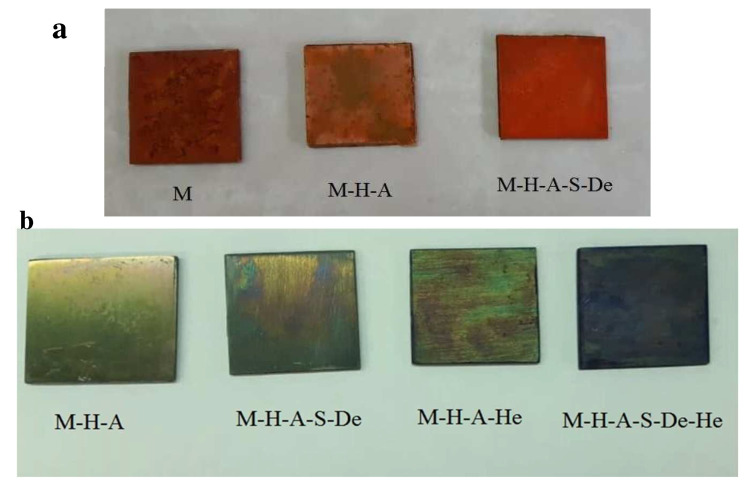
(**a**) Images of various samples after 5 h immersion in acid orange 7 (AO7) dye solution at 37 °C. (**b**) Images of samples following a 5 h immersion in a methylene blue dye solution at 37 °C. Samples were prepared with dimensions of 25 × 25 mm.

**Figure 4 jfb-13-00296-f004:**
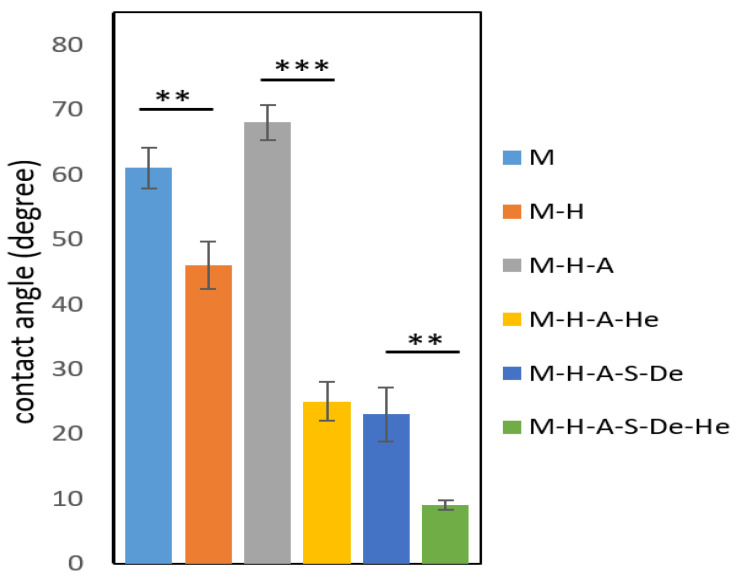
Static contact angle with water for different samples (** *p* < 0.05 and *** *p* < 0.001).

**Figure 5 jfb-13-00296-f005:**
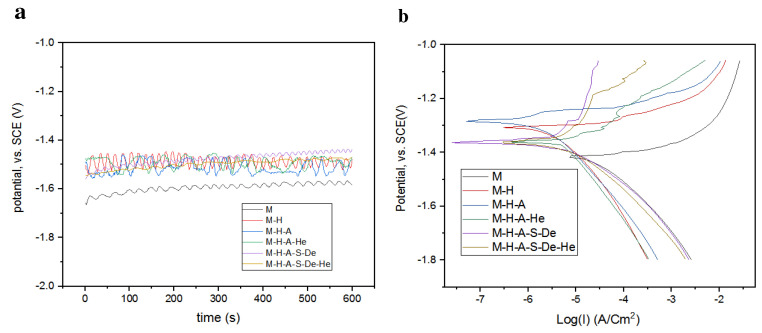
(**a**) Open circuit potential (OCP plots) and (**b**) potentiodynamic polarization plots (Tafel plots) for different samples in Hank’s solution.

**Figure 6 jfb-13-00296-f006:**
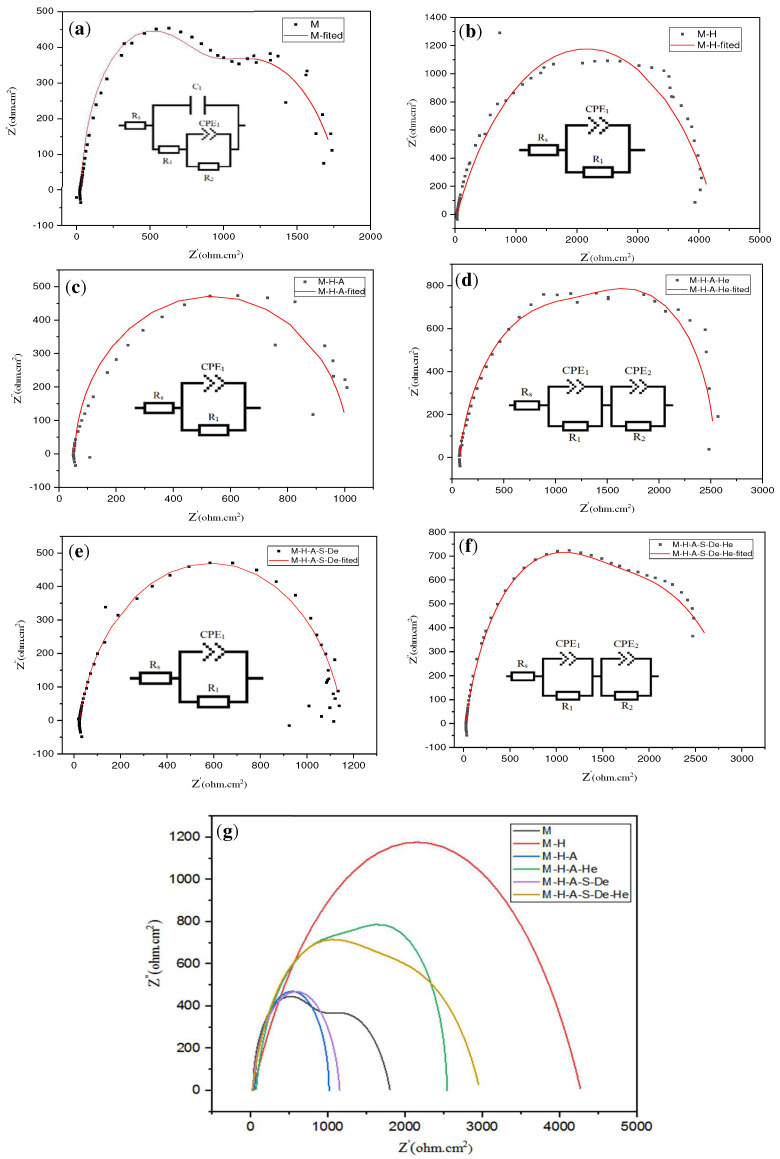
Nyquist curves for the samples (experimental, fitted data obtained with EIS analyzer software) and fitted equivalent circuits for each curve: (**a**) M, **(b**) M-H, (**c**) M-H-A, (**d**) M-H-A-He, (**e**) M-H-A-S-De, (**f**) M-H-A-S-De-He, and (**g**) simulated curve for different samples.

**Figure 7 jfb-13-00296-f007:**
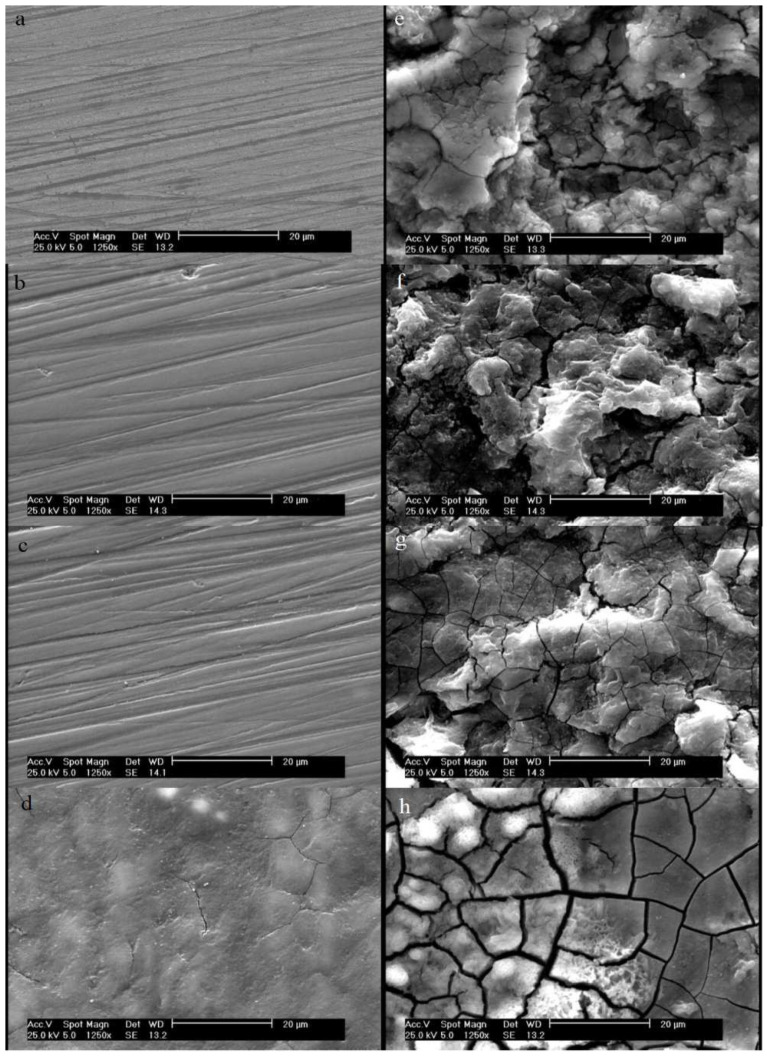
SEM micrographs obtained with ×1250 magnification: before electrochemical tests: (**a**) M, (**b**) M-H, (**c**) M-H-A, and (**d**) M-H-A-S-De; and after electrochemical tests: (**e**) M, (**f**) M-H, (**g**) M-H-A, and (**h**) M-H-S-De. Scale bar represents 20 µm.

**Figure 8 jfb-13-00296-f008:**
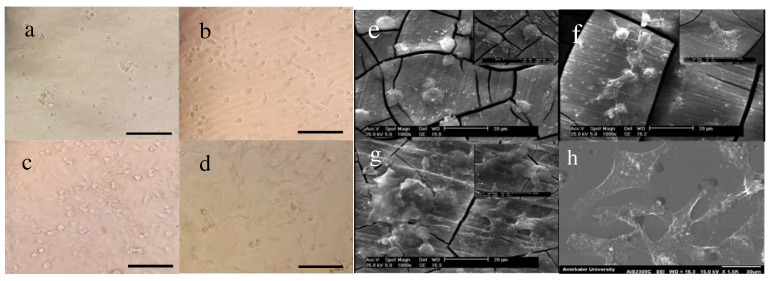
Optical micrographs of the HUVECs: (**a**) M, (**b**) M-H-A-He, (**c**) M-H-A-S-De-He, and (**d**) control. Scale bar represents 550 µm. SEM micrographs of the HUVEC cells: (**e**) M, (**f**) M-H-A-He, (**g**) M-H-A-S-De-He, and (**h**) control. Scale bar represents 20 µm.

**Figure 9 jfb-13-00296-f009:**
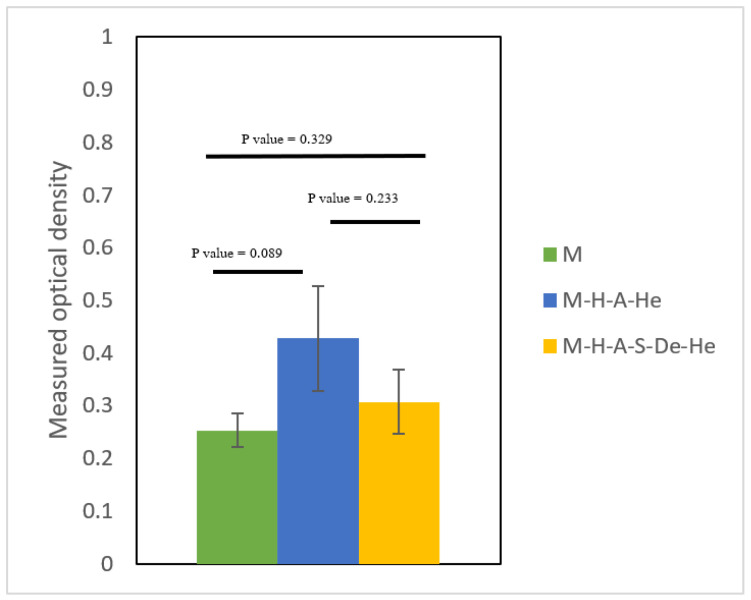
After 24 h of cell culture, HUVEC vitalities were quantified as percentages of the viabilities of cells in the control and heparin-modified AZ31 extractions (*p* > 0.05).

**Table 1 jfb-13-00296-t001:** Sample codes, types of coatings, and surface functional groups.

Sample Code	Definition	Top Coating	Surface Functional Groups
M	Unmodified Mg alloy	None	---
M-H	Hydroxidized Mg alloy	Mg(OH)_2_	OH
M-H-A	APTES-coated hydroxidized Mg alloy	APTES	NH_2_
M-H-A-He	Heparinized APTES-coated hydroxidized Mg alloy	Heparin	---
M-H-A-S-De	Dendrimers (PAMAM) grafted onto APTES-coated hydroxidized Mg alloy	PAMAM	NH_2_
M-H-A-S-De-He	Heparinized dendrimers (PAMAM) grafted onto APTES-coated hydroxidized Mg alloy	Heparin	---

**Table 2 jfb-13-00296-t002:** Wavelengths of FTIR peaks analyzed [22].

Sample	Peak Wavelength (cm^−1^)	Bonding
M	−	−
M-H	3684	O–H
M-H-A	1030, 1150	O–Si
	1556	N–H
M-H-A-He	1613	C=O
	1240, 1164	S=O, S–O
M-H-A-S-De	1563	N–H
	1613	C=O
	3000–3500	O–H
M-H-A-S-De-He	1168, 1240	S=O, S–O

**Table 3 jfb-13-00296-t003:** Acid Orange 7 (AO7) and methylene blue adsorption test results and densities of surface amine groups and heparin densities for different samples.

Sample	M	M-H-A	M-H-A-S-De	M-H-A-He	M-H-A-S-De-He	Blank
Acid Orange 7 optical density (OD)	1.168 ± 0.02	1.267 ± 0.07	0.877 ± 0.06	-	-	1.595 ± 0.01
NH_2_ density	4.3 × 10^−7^	3.3 × 10^−7^	7.2 × 10^−7^	-	-	−
Methylene blue optical density (OD)	-	1.23 ± 0.02	1.24 ± 0.05	1.18 ± 0.09	1.16 ± 0.07	1.24 ± 0.02
Heparin density	-	1.04 × 10^−7^	2.41 × 10^−8^	5.23 × 10^−7^	6.52 × 10^−7^	−

**Table 4 jfb-13-00296-t004:** Corrosion currents and corrosion potentials obtained from the PDP test and the results obtained with EIS analyzer software for different samples.

Sample	M	M-H	M-H-A	M-H-A-S-De	M-H-A-He	M-H-A-S-De-He
I_corr_ (µA)	25 ± 1.2	3.7 ± 0.7	0.62 ± 1.08	0.53 ± 4.96	0.014 ± 0.50	5.55 ± 0.37
E_corr_ (mV)	−1420 ± 23	−1300 ± 19	−1250 ± 14	−1350 ± 25	−1360 ± 15	−1350 ± 6
CPE_2_	1.5 × 10^−4^	−	0−	−	4.46 × 10^−6^	2.6 × 10^−4^
n_2_	0.7	−	−	−	1	0.83
Q_2_	2.00 × 10^−6^	−	−	−	4.46 × 10^−6^	3.20 × 10^−5^
R_2_	944.2	−	−	−	1100	1284.7
CPE_1_	6.30 × 10^−6^	5.1 × 10^−5^	3.14 × 10^−7^	7.90 × 10^−6^	4.4 × 10^−6^	2.0 × 10^−4^
n_1_	1	0.64	0.98	0.88	0.85	0.65
Q_1_	6.30 × 10^−6^	2.05 × 10^−5^	3.73 × 10^−7^	1.49 × 10^−5^	1.34 × 10^−6^	3.00 × 10^−4^
R_1_	829.5	4240	960.4	1130	1371.6	1663.4
R_s_	33.22	33.2	55.6	24.38	71.9	20.5

**Table 5 jfb-13-00296-t005:** Times of fibrin filament formation on the surfaces of the different samples.

Sample	M	M-H	M-H-A	M-H-A-He	M-H-A-S-De	M-H-A-S-De-He
Clotting time (s)	480 ± 10	445 ± 7	530 ± 12	630 ± 8	585 ± 11	715 ± 18

## Data Availability

Not applicable.

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
