# Peer review of "Surface Heparinization of a Magnesium-Based Alloy: A Comparison Study of Aminopropyltriethoxysilane (APTES) and Polyamidoamine (PAMAM) Dendrimers"

_jfb, 2022, doi:10.3390/jfb13040296_

Round 1
Reviewer 1 Report
Comments to authors are listed below:
1. The abstract should be improved by including some significant findings
2. The novelty and applications of this work should be reported clearly in the end of introduction section.
3. Regarding the discussions of results, they are very brief and did not cover all findings reported in detail.
4. FTIR spectra are not clear and should be re-drawn with high resolution figure.
5. SEM images in Figure 6, should be presented with high resolution magnifications.
Author Response
Reviewer 1
- The abstract should be improved by including some significant findings.
Answer: Thanks for your comment. More information was added to the abstract as follow:
…The presence of a peak with a wavelength of about 1560 cm-1 in the FTIR spectrum of the sample that was modified with APTES and dendrimer indicates the aminolysis of the surface…
… The results obtained from the potentiodynamic polarization (PDP) test showed that the corrosion current in the uncoated sample decreased from 25 µA to 3.7 µA in the alkali-heat treated sample.
- The novelty and applications of this work should be reported clearly in the end of introduction section.
The novelty of this work is the application of PAMAM dendrimers for surface aminolization of Mg-based stents. A comparison was made between the PAMAM dendrimers and APTES in terms of heparin grafting density on the surface of the Mg alloy It is mentioned like this at the end of the introduction:
The aim of this study is to investigate the effect of PAMAM dendrimers on surface aminolization of the AZ31 alloy. The resulting heparanized surface was compared with the APTES-aminolized surfaceson both corrosion resistance and blood compatibility [10, 23]. Since PAMAM dendrimers have more NH2 functional groups on their molecular surface and higher drug loading capacity than silanes (present in the APTES structure), they were effectively employed for the surface aminolization. The PAMAM structureleads to the entrapment of drugs between its branches. These two characteristics improve and increase heparinization. Additionally, dendrimer might offer greater corrosion resistance than silane due to the fact that its geometric and chemical structure serves as a corrosion inhibitor. To best of our knowledge, this is the first work to compare the corrosion resistance and blood compatibility of heparinized layer using both PAMAM and APTES.
- Regarding the discussions of results, they are very brief and did not cover all findings reported in detail.
Answer: Thanks for your comment. More details were added to the 3.1, 3.2, 3.4 and 3.5 sections.
- FTIR spectra are not clear and should be re-drawn with high resolution figure.
Answer: In the revised manuscript, the FTIR Figure was presented with higher quality as below:
- SEM images in Figure 6, should be presented with high resolution magnifications.
Answer: We would like to thank the reviewer for bringing this to our attention. The magnification of the SEM pictures is 1250 and scale bar represents 20 µm. Photos with better resolution were presented, as below:

Reviewer 2 Report
Dear authors!
thanks for the interesting article, but there are comments:
1. Figure 2 - image quality:
- the background must be white everywhere;
- it is necessary to increase the contrast of numbers on the graphs;
- make chart captions clearer and larger.
2. Figure 2b - gray column - measurement error is non-linear.
3. Figure 6 h - cracking of the coating is very strong. Will there be chipping of the coating? What is the adhesion of the coating?
4. Figure 7 - the letters g and h are missing from the images.
5. Figure 8 - There is no visible difference between the samples. I agree that more research is needed. Maybe this article should not provide this data?
6. Materials and methods, 2.7. For a wide audience of the journal there is no description: is the rate of blood clotting normal? A high clotting rate, just like a low clotting rate, is not always good for the human body.
Best regards!
Author Response
- Figure 2 - image quality, - the background must be white everywhere; it is necessary to increase the contrast of numbers on the graphs; make chart captions clearer and larger.
Answer: Thank you for your suggestion. The Figures were presented with higher quality in the revised manuscript.
- Figure 2b - gray column - measurement error is non-linear.
Answer: We would like to thank the author to draw this issue to our attention. Error bar data should not be linear. Initially we had drawn the graph using Origin software. In the revised manuscript, we draw the graph using Excel in which the measured error is presented correctly.
- Figure 6 h - cracking of the coating is very strong. Will there be chipping of the coating? What is the adhesion of the coating?
Answer: Figure 6 (h) despicts the surface of the sample after the completion of the corrosion tests. The existing cracks were caused by the corrosion of the surface, which is associated with the biodegradability of the coating. As seen in Figure 6 (d), the sample modified with PAMAM dendrimer before the corrosion test shows no crack on its surface. More explanation was added in the 3.2 section of the revised manuscript as follows:
Moreover, cracks and surface damage were not seen on the samples before the electrochemical test, indicating the stability of the coating. SEM images after the corrosion tests (Fig. 6(e- h)) qualitatively show the intensity of the surface degradation, which confirms the quantitative results obtained from the PDP test.
- Figure 7 - the letters g and h are missing from the images.
Answer: The comment is highly appreciated. The missing letters are given in the revised manuscript.
- Figure 8 - There is no visible difference between the samples. I agree that more research is needed. Maybe this article should not provide this data?
Yes, we also explained this issue in section 3.4 as follows:
The statistical results show that there is no significant difference between the cell viability for cells which were in touch with 24 h extracts of modified samples and the unmodified ones (P value > 0.05). It may be due to the short extraction time, and further studies in longer extraction times are suggested.
The data were expressed in a way that no toxicity was found in the samples due to the coating of the samples.
- Materials and methods, 2.7. For a wide audience of the journal there is no description: is the rate of blood clotting normal? A high clotting rate, just like a low clotting rate, is not always good for the human body.
The clot time assay test used in this study is only a comparison technique and a verification for the efficacy of heparin following surface grafting. It is able to compare the clotting times in the samples and look at the impact of coating and heparinization because all of the samples were examined under identical circumstances. Low clotting rates may not always be beneficial for the human body, but given the use of this coating (for cardiovascular stents to avoid early thrombosis and re-clogging), they may be more appropriate. More explanation is added in to the 3.5 section as follows:
Faster blood clot formation time shows the higher thrombogenic capacity of the material, which can cause in-stent restenosis in the implanted cardiovascular stents.
Surface modification by APTES and PAMAM has also contributed to reducing the time required for clotting. It has been reported in the literature that an amine functional group with positive charge can accelerate the time of clot formation [36]…

Round 2
Reviewer 1 Report
Comments to authors are listed below:
1. The results still should be discussed in detail because many significant findings presented in the discussion section are still missing or incomplete.
2. It is better that Figure 2 should be presented individually.
3. Conclusions should be re-written to reflect the size of results reported in this manuscript
Author Response
- The results still should be discussed in detail because many significant findings presented in the discussion section are still missing or incomplete.
Answer: Thanks for your comment. More details were added to different sections of results and discussions and related changes in the text are shown by track changes.
- It is better that Figure 2 should be presented individually.
Answer: In the revised manuscript, parts a and b of the figure 2 were separated and placed in two figure, and the number of the figures was also changed for this reason.
- Conclusions should be re-written to reflect the size of results reported in this manuscript.
Answer: Thanks for your comment. More details were added to the conclusion section as below:
…The aim of this study is to use PAMAM in order to increase the amine functional groups and increase the amount of heparin. FTIR test also confirmed the surface modification of the samples at each stage. …
… The formation of a passive hydroxide layer on the sample has a great effect on reducing the corrosion current and increasing the corrosion resistance. ….
… SEM images before and after the corrosion tests also supported the increase in corrosion resistance. …
